# PQ-VAE: Learning Hierarchical Discrete Representations with Progressive Quantization

## Abstract

Variational auto-encoders (VAEs) are widely used in generative modeling and representation learning, with applications ranging from image generation to data compression. However, conventional VAEs face challenges in balancing the trade-off between compactness and informativeness of the learned latent codes. In this work, we propose Progressive Quantization VAE (PQ-VAE), which aims to learn a progressive sequential structure for data representation that maximizes the mutual information between the latent representations and the original data in a limited description length. The resulting representations provide a global, compact, and hierarchical understanding of the data semantics, making it suitable for high-level tasks while achieving high compression rates. The proposed model offers an effective solution for generative modeling and data compression while enabling improved performance in high-level tasks such as image understanding and generation.

## 1 Introduction

Variational auto-encoders (VAEs) (Kingma & Welling, 2013) are powerful tools for generative modeling and learning data efficient representations, with applications in diverse fields such as image generation, anomaly detection, and data compression. VAEs optimize the Evidence Lower Bound (ELBO) objective function that includes two terms, a reconstruction loss and a regularization term (KL divergence), encouraging the latent variables to follow a prior distribution. These two terms balance the complexity of the model and the amount of information needed to describe the input data, following the Minimum Description Length (MDL) principle (Rissanen, 1978). The reconstruction loss and the regularization term can be seen as measures of the information in the latent code and the information required to recover the input.

We want the latent codes to be compact, meaningful, and represented by as few bits as possible. On the other hand, it is also desirable that the latent codes provide as rich information as possible, which requires the model to be confident about the latent codes given the input. These two goals are conflicting in the ELBO. In conventional VAE objectives, a tradeoff between these two goals can be adjusted by the choice of a hyperparameter known as the beta coefficient. However, the effectiveness of this approach can depend on the particular application and choice of beta value. Furthermore, earlier works (Alemi et al., 2018) have shown that a tighter ELBO does not necessarily lead to better latent representations. To address these limitations, several extensions to the VAE framework have been proposed, such as the beta-VAE (Higgins et al., 2016) and total correlation (TC)-VAE (Chen et al., 2018), which aim to better balance the tradeoff between description length and information content in the latent code. However, the use of continuous latent codes in these works does not effectively reflect the description length of the codes. For example, to use them for image compression, additional steps such as scalar quantization and entropy coding are necessary, and this is not modeled as part of the original formulation.

Vector Quantized VAE (VQ-VAE) Van Den Oord et al. (2017) is a successful discrete VAE model that uses vector quantization to discretize the continuous latent variables and achieve high compression rates, generating compact and meaningful codebooks. Its variants, such as VQ-VAE2 (Razavi et al., 2019), VQ-GAN (Esser et al., 2021), and Vit-VQGAN (Yu et al., 2021), use hierarchical structures, GAN training, or transformer structures (Vaswani et al.) to further improve reconstruction quality. However, when considering images as the data, VQ-VAE only learns small 2D tokens associated

with local regions of the original image. Modeling the 2D structure can be challenging, even on a smaller scale, which requires training a heavy-weighted prior model like PixelCNN or transformers to generate images. Furthermore, since high-level tasks such as image understanding require a global understanding of the image's semantics, VQ-VAE learns local and low-level representations that may not be suitable for such tasks.

In this work, we propose to learn a progressive sequential structure of data representations, aiming to maximize the information content in a limited (code) length. Our approach results in global, compact, and hierarchical representations, with information organized based on its relevance to the given data. The advantages of this proposed progressive structure are manifold: 1) considerable compression rate can be achieved by encouraging the representations to be compact; 2) the representations are easier to understand, *e.g.*, for image classification, and the most important information can be easily extracted by taking the beginning part of the learned latent codes; 3) they are easier to model, for generation tasks where auto-regressive methods are prevalent, *e.g.*, image generation, the progressive structure is more natural than raster scan because the order of the latent codes inherently reflects the level of information.

Our contributions can be summarized as:

1. We propose to learn a hierarchical sequence of discrete representations.

2. We propose progressive quantization and leverage information maximization to obtain such hierarchical sequential representations.

3. Extensive experiments on image reconstruction, image generation demonstrate the superiority to other VQ-based methods.

4. Qualitative analysis shows that the learned latent codes exhibit a hierarchical structure.

## 2 RELATED WORK

### 2.1 REPRESENTATION LEARNING

Representation learning focuses on learning representations of data that are useful for downstream tasks. These representations are typically learned through unsupervised learning methods, which do not require explicit supervision or labels.

Classic works in representation learning include Principal Component Analysis (PCA), which finds a linear subspace that captures the most variance in the data; Independent Component Analysis (ICA), which seeks to uncover independent sources that can account for the observed data; K-means, which automatically groups data into different clusters; autoencoders, which convert data into lower-dimensional representations.

Other notable works in representation learning include Contrastive Predictive Coding (CPC) (Van den Oord et al., 2018), which learns representations by predicting latent tokens in an auto-regressive manner; and the InfoNCE (Van den Oord et al., 2018) objective, which learns representations by maximizing the mutual information between the input and the latent context.

Recent advances in unsupervised representation learning have focused on learning representations that capture more complex and abstract features of the data. Contrastive learning has emerged as a powerful approach for this task, where the model learns to discriminate between similar and dissimilar pairs of data points in the latent space. This approach has been used in recent works such as InfoMin (Tian et al., 2020) and SimCLR (Chen et al., 2020) to learn highly informative and transferable representations that outperform previous state-of-the-art methods on various downstream tasks. Another line of recent state-of-the-art methods are masked modeling. BERT (Devlin et al., 2018), GPT (Radford et al., 2018), and their variants (Liu et al., 2019; Radford et al., 2019; Brown et al., 2020) are successful pre-training methods in NLP. They mask out some portion of the input text and use the rest to reconstruct the missing words. MAE (He et al., 2022) extended this method to image data and showed success in image pre-training.

## 2.2 Variational Autoencoders

Variational autoencoders (VAEs) have become a popular approach for unsupervised representation learning. Various methods have been proposed to improve the original VAEs, including Importance Weighted Autoencoder (IWAE) (Burda et al., 2015) which uses importance sampling to improve the tightness of the lower bound, and VAE with Inverse Autoregressive Flow (VAE-IAF) (Kingma et al., 2016) which replaces the simple Gaussian posterior approximation with a more flexible model.

Another line of research focuses on improving the disentanglement ability of VAEs. Beta-VAE (Higgins et al., 2016) proposed a modification to the VAE objective function by adding a hyperparameter beta, which controls the trade-off between the reconstruction loss and the KL divergence term in the Evidence Lower Bound (ELBO) objective function. FactorVAE (Kim & Mnih, 2018) and beta-TC VAE (Chen et al., 2018) introduce additional terms in the loss function to encourage independence between different latent dimensions. Wasserstein Autoencoder (WAE) (Tolstikhin et al., 2017) uses a Wasserstein distance instead of KL divergence as the regularization term. InfoVAE employs information-theoretic principles to guide the learning of a more disentangled representation.

Finally, there is a growing interest in using VAEs for discrete data. Vector Quantized VAE (VQ-VAE) (Van Den Oord et al., 2017) employs discrete latent variables and a codebook to compress the input data, while VQGAN (Esser et al., 2021) applies additional GAN training to make the reconstrucion more realistic to achieve higher compression rates. These methods have shown promising results in image compression and generative modeling.

The proposed method is related to recent advances in using VAEs for unsupervised representation learning. However, it differs by introducing a progressive structure of latent representations. This unique approach allows us to simultaneously minimize of description length and maximize the information content within the latent codes of VAEs. As a result, our method generates representations that are not only compact but also meaningful.

## 3 Background

### 3.1 VAE

Variational Autoencoders (VAEs) are a type of neural network used for generative modeling. The VAE consists of two main parts, an encoder and a decoder. The encoder maps the input data, such as an image, to a latent space, while the decoder maps the latent vectors back to the original data space. The encoder can be represented as a function $q_\phi(z|x)$ modeled by parameters $\phi$ that maps the input data $x$ to a distribution over the latent space $z$. The decoder can be represented as a function $p_\theta(x|z)$ modeled by parameters $\theta$ that maps the latent vectors $z$ to a distribution over the original data space $x$.

To train the VAE, we aim to maximize the likelihood of the observed data given the latent variable $z$ using a variational approximation $q(z|x)$, which is typically chosen to be a simpler distribution such as a multivariate Gaussian. The objective function for the VAE can be written as,

$$\log p_\theta(x) \geq \mathcal{L}(\theta, \phi) = \mathbb{E}_{q_\phi(z|x)} \left[ \log p_\theta(x|z) \right] - D_{KL}(q_\phi(z|x)||p(z)), \tag{1}$$

where $\mathbb{E}$ denotes the expectation over the latent variable $z$, and $D_{KL}$ is the Kullback-Leibler divergence between the variational approximation and the prior distribution $p(z)$. The first term in the objective function encourages the reconstructed data to be similar to the original input, while the second term regularizes the latent distribution to be close to the prior distribution.

### 3.2 VQ-VAE

Let $x$ be the input data, $z$ the continuous latent variable, and $e(x)$ the encoder function that maps $x$ to $z$. The vector quantization module maps the continuous latent variable $z$ to a discrete code $q(z)$ from the codebook $C$. The decoder function $d(z)$ maps the latent variable $z$ to the output space, such that $d(q(z))$ approximates $x$. The loss function for VQ-VAE is a combination of a reconstruction loss, a codebook loss, and a commitment loss, given as,

$$\mathcal{L} = \mathbb{E}_{x \sim p_{data}} \left[ |x - d(q(e(x)))|^2 \right] + \beta|\text{sg}(z) - q(z)|^2 + \gamma|z - \text{sg}(q(z))|^2,$$

where $x_q$ is the reconstruction of $x$ using the code $q(z_e)$ and $\beta$ and $\gamma$ are hyperparameters that control the weighting of the codebook loss and the commitment loss, respectively, and $\text{sg}(\cdot)$ denotes the "stop gradient" operation.

The codebook loss term encourages the discrete codes to represent the input data well, and the commitment loss term encourages the encoder to commit to a code in the codebook for each input. The use of discrete codes enables VQ-VAE to learn compact and meaningful representations of the input data.

---

**Algorithm 1** Progressive Quantization
___
 1: **Input:**
 2: $F$ : Feature tensor of shape $(c, H, W)$.
 3: **Output:**
 4: $Z_Q$ : Simplified quantized feature tensor.
 5: **Codes** : Index tensor representing the selected codes from the codebook.
 6: **procedure** PROGRESSIVE QUANTIZER
 7:     Prepare input feature map $F$ and group feature vectors.
 8:     **for** each latent $i$ in range(n_steps) **do**
 9:         Apply step-specific transformations to obtain transformed feature.
10:         Sample the codebook entries with Gumbel-Softmax.
11:     **end for**
12:     Extract $Z_Q$ as a succinct representation of $F$.
13:     **return** $Z_Q$ and **Codes**
14: **end procedure**

---

## 4 PROPOSED APPROACH

Our goal is to maximize the informativeness of limited latent codes. On the one hand, the reconstruction loss in the ELBO objective reflects the information attained from the input; on the other hand, we can also measure the mutual information between the data and the latent codes, which is

$$I_q(x, z) = H(q(z)) - \mathbb{E}_{x \sim p_{data}(x)} H(q(z|x)), \tag{2}$$

*i.e.*, the entropy of the latent code distribution minus the conditional entropy given the input data. Next we will present in detail in Section 4.1 our formulation of a new training loss that maximizes information in limited discrete codes. In Section 4.2, we will further show that, guided by the proposed training objective, a meticulously designed progressive vector quantization procedure introduces a hierarchical sequential structure into representations.

### 4.1 MAXIMUM INFORMATION IN LIMITED DISCRETE CODES

We assume the unobserved random variable $z$ for generating data point $x$ is a sequence of $L$ discrete tokens, meaning $z \in \{(z_1, z_2, \ldots, z_L) \mid z_1, z_2, \ldots, z_l \in \{1, 2, \ldots, K\}\}$, where $K$ denotes the size of the vocabulary. The sequence $\{z_1, z_{1:2}, \ldots, z_{1:L}\}$ inherently exhibits a hierarchical structure, as each subsequent term must incorporate its preceding term at the forefront of its representation. In addition, we assume the tokens are drawn independently from a uniform distribution, $P(z_1, z_2, \ldots, z_L) = \prod_{n=1,\ldots,L} P(z_n) = \frac{1}{K^L}$. Thus,

$$\text{KL}\left(q(z \mid x) \| p(z)\right) = L \log K - \sum_{l=1}^{L} H(q(z_l \mid x, z_{<l})). \tag{3}$$

The KL divergence between the posterior and the prior distribution, which reflects the rate needed to describe the latent code, is negatively related to the entropy of the posterior distribution. It is typically included in the conventional continuous VAE training objective so that the required number of bits is not infinity because the KL divergence between two continuous distributions is not upper-bounded. However, in the discrete settings, the KL is upper bounded. The upper bound is achieved when the conditional entropy of $z_l$ is zero at all positions, meaning the model is absolutely confident about the choice of the latent code given the input. This is desired for meaningful representations. When

the conditional entropy becomes all zero, then the KL divergence achieves its upper bound and is constant. Therefore, we remove it from the training objective.

We want to constrain that the entropy of the approximate posterior is zero. Besides, we also need to constrain that the aggregated posterior distribution matches the prior, *i.e.*, $KL(q(z_{1:L})||p(z_{1:L})) = H(p(z)) - H(q(z)) = 0$. Therefore, we add a regularization term, the negative mutual information between the data and the latent codes, to the training loss. Finally, the proposed training loss becomes,

$$\mathcal{L} = \left(\mathbb{E}_{z \sim \hat{q}(z_{1:L}|x)} \log p(x|z)\right) + \lambda_h \sum_{l=1}^{L} \left(H(q(z_l \mid x, z_{<l})) - H(q(z_l|z_{<l}))\right). \tag{4}$$

It is non-trivial to estimate the aggregated posterior distribution $q(z)$ as it requires enumerating the whole dataset. Even if the estimation is accurate enough, when $H(q(z|x))$ is minimized for each $x$ in the dataset, $H(q(z))$ will not exceed $\log N$, where $N$ is the size of the dataset. We notice that in some works $H(q(z_{l_a} \mid x)) - H(q(z))$ can be further decomposed as a total correlation (TC) term plus a dimension wise KL term, and emphasize the TC term. Here we care about compactness over independence. In fact, if $H(q(z|l))$ is maximized for some $l$, then it means that $q(z|l) = p(z|l)$ and both the TC term and the dimension wise KL will be minimized. Therefore, we will optimize the expected entropy solely. Therefore, we relax the formulation of $H(q(z_l \mid x, z_{<l})) \approx H(q(z_l \mid x))$, which can be estimated easily using Monte Carlo sampling within a minibatch.

## 4.2 Progressive Vector Quantization

Standard VAEs convert the encoder's feature map of the input image to a feature vector, whereas VQ-VAEs quantize each feature vector. In our approach, we convert the feature map to a feature vector $\mathbf{z}$ and quantize it progressively, while allowing for flexibility in the size of the latent codes.

The Progressive Quantization technique is meticulously designed, leveraging a codebook and position-specific reversible transformations to quantize input feature maps efficiently and effectively. The process unfolds through several distinct, systematic steps, elucidated below:

**Input Feature Map Preparation:** Given an input feature map characterized by the shape $(c, H, W)$, it is initially partitioned into $h \times w$ non-overlapping patches, each delineated by the shape $(H/h, W/w)$ (we assume H, W can be divided even by h, w for simplicity).

**Feature Vector Grouping:** Following the partitioning, feature vectors are grouped based on their relative positions within each respective patch, yielding a reshaped feature map of shape $(H/h \cdot W/w, c, h, w)$. This grouping scheme ensures that each obtained code possesses an equally nearly full-size receptive field, thereby encapsulating a comprehensive, global overview of the original feature map.

**Progressive Quantization:** Subsequent to the reshaping, the feature map is subjected to a sequence of quantization steps, each entailing a unique, step-specific transformation. These transformations are composed of two linear layers, acting on the spatial and channel dimensions respectively. An transformed feature is derived and flattened. Distances between this transformed feature and the codebook entries are then computed, with the Gumbel-Softmax technique being utilized to sample the codebook entries. Following this, a reversed transformation is applied, utilizing a pseudo-inverse weight matrix, and the resultant map is subtracted from the reshaped feature map. This procedure is reiterated until the predetermined maximum number of quantization steps is attained.

**Quantized Feature Map Extraction:** Upon the culmination of all the quantization steps, the refined quantized feature map is extracted. This map is a succinct and effective representation of the initial input feature map.

This progressive quantization procedure offers a structured and precise methodology for feature map quantization. It leverages position-specific reversible transformations, enhancing the accuracy and representativeness of the quantized map. The careful consideration of the receptive field in the feature vector grouping stage ensures the encapsulation of global information. The mutual information term in Equation 4 encourages each latent code at each quantization step to attain as much information as possible, thus resulting in a hierarchical sequential structure of representations.

Table 1: Source images from MNIST, CelebA and LSUN Church datasets and reconstructions with different models. All models use the same number of discrete latent codes for each dataset.

| Methods | MNIST | CelebA | LSUN Church |
|---------|-------|--------|-------------|
| Source |  |  |  |
| VQ-VAE |  |  |  |
| SQ-VAE |  |  |  |
| VQ-WAE |  | - | - |
| PQ-VAE |  |  |  |

## 5 EXPERIMENTS

We perform empirical evaluations of the proposed method using the MNIST, CIFAR-10 Liu et al. (2018), CelebA and LSUN Church datasets. We measure the rate-distortion to assess its compression ability. Qualitative experimental results are also presented to investigate the generation process and illustrate the model's ability to learn a hierarchical structure of latent representations. Additional results can be found in Appendix.

### 5.1 RECONSTRUCTION

We show the test images from MNIST, CelebA and LSUN Church and the reconstructed images from our PQ-VAE model in Table 1. For comparison, various other models including VQ-VAE, SQ-VAE (Takida et al., 2022), VQ-WAE (Vuong et al., 2023) are also listed. For all methods, we use the same number of latent codes and codebooks of the same size. Specifically, the codebook size is set to 256 for MNIST and 256 for other datasets; the number of latent codes are 64 for MNIST and CIFAR10, 256 for CelebA, and 1024 for LSUN Church. For VQ-VAE, SQ-VAE, VQ-WAE, MNIST images are resized to 32x32 compressed to 8x8; CelebA images are performed a 140x140 centercrop, resized to 64x64 and then compressed to 16x16; LSUN Church images are resized to 128x128 and compressed to 32x32. For PQ-VAE, we use 64, 256 and 1024 latent codes for the three datasets respectively. The number of latent codes of PQ-VAE are not necessarily equal to the feature map size but is selected so for fair comparisons.

For quantitative assessments, we report in Table 2 the mean square error (MSE) and the reconstruction Fréchet Inception Distance (rFID) between the test images and the reconstructed images from different models. VQ-WAE is not included for comparision on CelebA and LSUN as it took too much time to train. As we can see, SQ-VAE and VQ-WAE outperforms the original VQ-VAE by a large margin in terms of both MSE and rFID. These two methods and many recent VQ-VAE based methods improve the VQ-VAE method by addressing the codebook collapse issue, which is when the codebook usage is low, thus making the training sub-optimal. Our method diverges from this kind of improvement, and improves VQ-VAE from a different perspective through the establishment of hierarchical latent codes with progressive quantization. Even so, our method shows superior performances to SQ-VAE and VQ-WAE.

### 5.2 GENERATION

Like all other VQ-VAE methods, to generate images, we need to learn an additional prior network to model the distribution of the latent codes. It is noteworthy that our method does not require much specific prior knowledge about model architecture of the prior network. This is because PQ-VAE

Table 2: Mean Square Error (MSE) ($\times 10^3$) and reconstruction Frechlet Inception Distance (rFID) of models on MNIST, CIFAR10, CelebA and LSUN Church datasets.

| Methods | MNIST | | CIFAR10 | | CelebA | | LSUN Church | |
|---|---|---|---|---|---|---|---|---|
| | MSE | rFID | MSE | rFID | MSE | rFID | MSE | rFID |
| VQ-VAE | 0.78 | 3.32 | 3.63 | 77.30 | 1.32 | 19.4 | 1.84 | 73.53 |
| SQ-VAE | **0.42** | 2.73 | 3.27 | 55.40 | 0.96 | 14.8 | 1.79 | 70.26 |
| VQ-WAE | 0.51 | 1.67 | 3.43 | **54.30** | 1.05 | **14.2** | - | - |
| PQ-VAE | 0.48 | **1.51** | **2.96** | 65.06 | **0.82** | 22.78 | **1.49** | **69.98** |

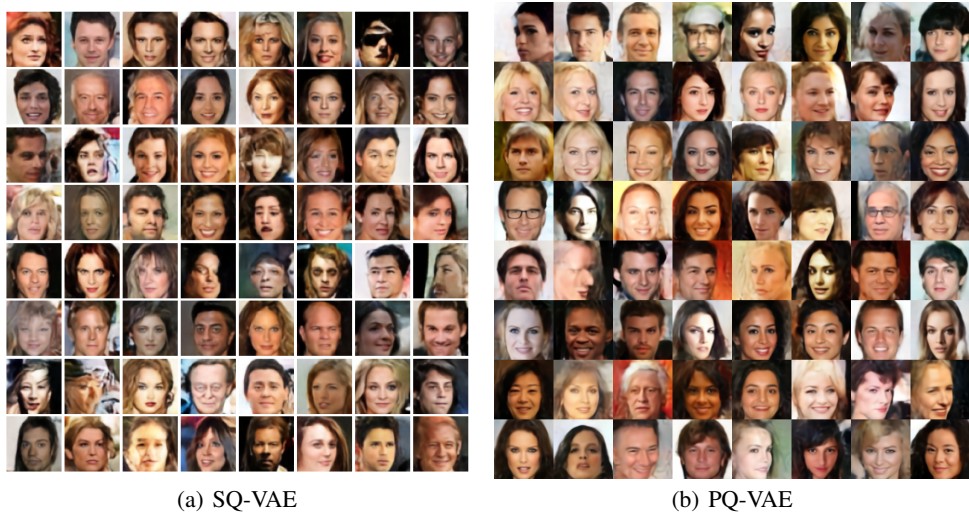

(a) SQ-VAE          (b) PQ-VAE

Figure 1: Generated samples of CelebA using SQ-VAE and PQ-VAE.

produces codes that has a hierarchy. And the latent codes in other methods presents a 2-dimensional spatial structure. Thus, CNN like models are needed for them such as PixelCNN, PixelSNAIL. In contrast, the latent codes of PQ-VAE is a 1-dimensional sequence and is ordered by the amount of information they possess. We can utilize sequence models for learning prior distributions. Specifically, we use a transformer to model the conditional probability of latter codes given earlier codes. For training the transformer network, apart from adding a position-specific embedding to the input, we also apply a position-specific linear transformation, which corresponds to the progressive quantization process.

We perform image generation of our method and the baseline SQ-VAE method CelebA. We visulize the generated images in Figure 1. PQ-VAE generates much shaper images than others. Looking into the details of the generated images of CelebA, we can find that face images generated by PQ-VAE are more cohesive, while images generated by other methods can have misaligned eyes, noses, mouths and etc.

## 5.3 STUDYING THE LEARNED LATENT CODES

We conduct qualitative experiments to study the learned structure of the latent codes and their importance in image reconstruction and classification.

### 5.3.1 MEASURING THE AMOUNT OF INFORMATION IN LATENT CODES

To measure the information in a latent code, we can calculate the mutual information between the code and the input. The mutual information is not directly accessible but can be estimated in two

Table 3: Progressive reconstructions of the PQ-VAE model on MNIST and CelebA datasets. First $l$ latent codes are extracted from PQ-VAE model and the rest codes are sampled from the prior network given the first $l$ ones. The combined codes are decoded to reconstruct the source images.

ways by using its definition: $I(x; z) = H(x) - H(x|z) = H(z) - H(z|x)$. Firstly, we evaluated the importance of the latent codes at different positions by removing a latent code at a position and measuring the reconstruction MSE. Secondly, we compute the conditional entropy of a latent code given the source image and the previous codes. Figure 2 shows the results. It illustrates that removing a latent code at an early position results in a significantly larger MSE, while removing a later one does not make a big difference. This suggests that the latent codes are ordered by their importance for reconstructions, and a semantic hierarchical structure of latent codes is successfully learned. On the other hand, the conditional entropy tend to be growing as the progressive quantization goes, which also shows the amount of information is more at ealier positions. This also means the model is more

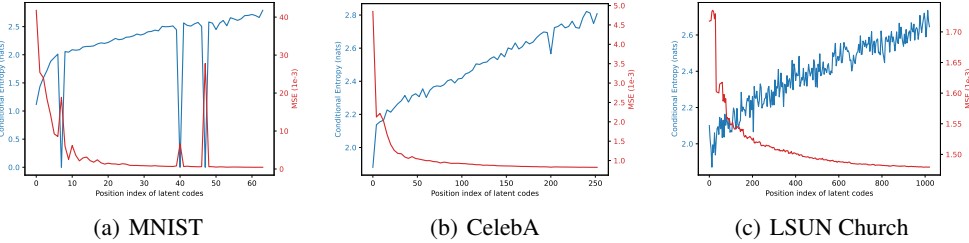

|  (a) MNIST | (b) CelebA | (c) LSUN Church |

Figure 2: Studies of the amount of information contained in latent codes at different positions. We plot MSE as a function of removed latent code index, and the conditional entropy of a latent code given the input image and the previous codes. This graph illustrates the hierarchical significance of latent codes at different positions, indicating that earlier latent codes carry more important information compared to the later ones.

confident at the beginning while more stochastic as it goes. The intuition is that semantic information, which is stored at the earlier part is more unique for a given image while the details (in the later part) are more random.

### 5.3.2 PROGRESSIVE RECONSTRUCTIONS

To understand what information the latent codes are possessing. We show the progressive image reconstructions of PQ-VAE in Table 3. We take the first $l$ latent codes according to the posterior distribution of the latent codes given the source images and randomly samples the rest according to the prior distribution learned from the prior network. As it shows, even at the beginning of the process when the length is small, PQ-VAE can generate sharp images, and the reconstructions are semantically similar to the original images. At the beginning, PQ-VAE can be confused between several digit pairs. As it uses more latent codes, it becomes more confident about the semantic information, and the label information is stable (from $l$=2 for MNIST). As the length of codes keeps increasing, fine details are added to the reconstruction.

## 6 CONCLUSION

Redundancy is a common feature in data, leading to increased processing costs and difficulties in interpretation. A compact and semantic representation of the data is generally considered to be more meaningful. While traditional compression methods rely on prior knowledge of the data, recent learning methods aim to model data distributions. In this work, we propose PQ-VAE that directly maximizes information given limited data description lengths in a semantic hierarchical fashion. Our extensive empirical results demonstrate that our approach achieves superior compression rates and captures semantic information of data.

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

## A    Implementation details

For VQ-VAE, SQ-VAE and VQ-WAE, we follow the description in Takida et al. (2022) and Vuong et al. (2023) to build models for all datasets except LSUN Church which is not included in their paper and we use the same model as in CelebA. The training details are also identical as in their settings.

For PQ-VAE, we adopt the encoder and decoder structures from dVAE (Ramesh et al., 2021) and use 3 ResNet blocks both for the encoder and the decoder. We transform the encoded feature to 512-dimensional . For the progressive quantization, we always partition the feature map into $8 \times 8$ patches, and use a weight matrix of size $64 \times 16$ to transform the spatial dimension and a weight matrix of size $512 \times 4$ to transform the channel dimension of the feature map.

We use an initial learning rate of $1e - 3$ which is anneal at training step $ts$ by $e^{\frac{-ts}{25000}}$. We use batch size 128 for MNIST, CIFAR; 512 for CelebA; and 128 for LSUN Church.

