# OpenReview forum: "PQ-VAE: Learning Hierarchical Discrete Representations with Progressive Quantization"
_ICLR.cc/2024/Conference — ICLR 2024 Conference Withdrawn Submission_

### Official Review · Reviewer_L7SW · 2023-10-17

**Soundness:** 1 poor
**Presentation:** 1 poor
**Contribution:** 1 poor
**Rating:** 1
**Confidence:** 4

**Summary:**

The paper presents a residual vector quantizer (if I understand correctly, the method is poorly described). Prior similar work is not included. The training loss is modified in a way that is supposed to increase the entropy of the approximate posterior to be zero (unclear why this is done).

**Strengths:**

N/A

**Weaknesses:**

I have a lot of problems with this paper.

Firstly, the method is very poorly described and not remotely clear until Sec 4.2 on p5, i.e., Introduction and Abstract are not telling the reader what's going on. In Sec 4.2 we then see the following: the method is simply doing residual vector quantization on h x w patches, calling this "progressive quantization": "[...] and the resultant map is subtracted from the reshaped feature map". Sadly, the two prominent prior works on residual quantization are ignored in this work, in a big blow to it's credibility and novelty:

Lee et al, Autoregressive Image Generation using Residual Quantization, https://arxiv.org/abs/2203.01941
Zeghidour et al, SoundSTream: An End-to-End NEural Audio Codec, https://ieeexplore.ieee.org/document/9625818

Overlooking this for the moment, we turn to Sec 4.1, which is very confusing indeed. The authors introduce z as a sequence of tokens which apparently already "inherently exhibits a hierarchical structure" although it's not clear why at this point in the text. The authors claim "we assume the tokens are drawn independently from a uniform distribution" which seems a very wild to me, and is unmotivated. Maybe the idea is that a typical VQ VAE is supposed to have uniform utilization? But of course the joint distribution is never uniform. At the end of p4, we have a weird mix between continuous VAEs and discrete VQ-VAEs (typically trained without an explicit prior): "[...] the model is absolutely confident about the choice of the latent code given the input. This is desired for meaningful representations." Are we reinventing VQ-VAEs here? I am very lost. As a minor gripe, on p5, we have an incorrect formula for KL (should be H(p,q) - H(q)).

At this point, I am very confused indeed what the method is, but let us turn to "Experiments". We first see better reconstruction metrics (Table 2), in line with the above RVQ work (that is, alas, not in any comparisons). In Sec. 5.2, the authors claim "the latent codes in other methods presents a 2-dimensional spatial structure. Thus, CNN like models are needed for them such as PixelCNN, PixelSNAIL." Clearly a false statement given the breath of literature that uses transformers directly on VQ VAEs (eg Esser et al, that is cited in this work). The authors continue "In contrast, the latent codes of PQ-VAE is a 1-dimensional sequence and is ordered by the amount of information they possess." This surprised me, since it was not previously clear that there would be an ordering. Finally, "We can utilize sequence models for learning prior distributions. Specifically, we use a transformer to model the conditional probability of latter codes given earlier codes". Now, we only see one visual figure for the result of the generation study, Fig 1, which only includes one baseline. It is unclear why we don't see FID, since that was included in Table 2. I don't think we can draw any conclusions from a single visual figure.

Finally, in Appendix A it seems like a different autoencoder was used for this paper. This is a bit surprising to me and makes me wonder if the comparisons are fair. In principle, the same VAE and transformer arch should be used for all quantizers.

**Questions:**

- How does PQ compare to RVQ methods?
- Can you rewrite Sec 4. to be more clear? A figure would help.

---

### Official Review · Reviewer_JP89 · 2023-10-31

**Soundness:** 2 fair
**Presentation:** 2 fair
**Contribution:** 2 fair
**Rating:** 3
**Confidence:** 4

**Summary:**

The authors propose Progressive Quantization VAE (PQ-VAE), which aims to learn a progressive sequential structure for data representation that maximizes the mutual information between the latent representations and the original data in a limited description length.

**Strengths:**

1. This proposed model shows improved reconstruction and generation quality in some datasets.
2. The author conduct experiments on multiple datasets and compared PQ-VAE with multiple baselines.

**Weaknesses:**

1. The authors' explanation of progressive quantization lacks clarity. I recommend supplementing the text with more diagrams or visual aids to elucidate how the quantization process operates.

2. Equation 4 appears to merely represent an autoregressive formulation of the discrete latent variable, which raises questions about the novelty of this concept. The authors might want to highlight the unique aspects of their approach to distinguish it from existing methods.

3. Table 2 presents the reconstruction qualities of various models, but lacks sufficient information about each model's configurations. Notably, the reconstruction quality is influenced by the $\lambda_h$ factor in Equation 4, suggesting that a lower $\lambda_h$ could enhance the reconstruction quality. For a more balanced evaluation of reconstruction quality, the authors should also consider the rate of the latent representation.

4. The model's performance seems somewhat restricted. While the Fréchet Inception Distance (FID) is typically employed to assess the perceptual quality of natural images, making it suitable for datasets like CelebA and LSUN Church, the Mean Squared Error (MSE) is more reliable for MNIST. However, the proposed model does not outperform the VQ-WAE.

5. Another concern with the FID is its status as a non-reference metric for perception. The authors should consider incorporating reference-based perceptual metrics such as the Learned Perceptual Image Patch Similarity (LPIPS) into their evaluation of reconstruction.

6. The model has only been tested on low-resolution images. The standard resolution for datasets like CelebA and LSUN Church is approximately 256x256, so it would be beneficial to evaluate the model on these higher-resolution images.

7. Regarding the generation results, the authors should consider providing quantitative results in addition to the qualitative samples. This would help readers gain a more comprehensive understanding of the model's performance.

**Questions:**

see above

---

### Official Review · Reviewer_bWca · 2023-11-01

**Soundness:** 2 fair
**Presentation:** 1 poor
**Contribution:** 3 good
**Rating:** 3
**Confidence:** 4

**Summary:**

This paper introduces Progressive Quantization VAE (PQ-VAE), a variational auto-encoder designed for generative modeling and representation learning. PQ-VAE addresses the balance between compactness and informativeness in latent codes, creating global, compact, and hierarchical data representations. These characteristics make it well-suited for high-level tasks and data compression, enhancing performance in applications like image understanding and generation.

The model is promising due to its unique approach to performing quantization on latent features. However, it has drawbacks including missing related works, a lack of formal definitions and visual aids, and an unclear demonstration of the importance of mutual information.

The paper could be improved by providing clear mathematical formulations, comparing the model's performance using rate-distortion theory, addressing code utilization issues, and clarifying the choice of Gumbel-softmax for the vector quantization bottleneck.

**Strengths:**

The future potential of this model seems high since it performs quantization on the latent features instead of on a feature map with spatial resolution.

**Weaknesses:**

1. Missing related works: The paper could benefit from a comparison with methods such as product quantization [2] and recursive quantization [3], which also perform quantization on latent features.

2. Poor presentation: The paper lacks formal mathematical definitions and an overall figure to describe the method.

3. The importance of mutual information is not demonstrated. The paper seems to contradict itself by first using mutual information as a tool to increase code utilization, and then showing that mutual information is maximized, which seems like a circular argument.

**Questions:**

1. Mathematical formulation: It would be beneficial to provide a clear mathematical definition of the method, drawing from examples in other research. The current explanation in the paper is not sufficient for a full understanding of the intended process.

2. Merits of the proposed method using rate-distortion theory: Does the proposed research offer advantages over existing vector quantization methods when viewed from the perspective of rate-distortion theory? In other words, does it provide a theoretical guarantee for better reconstruction performance when the codebook size and number of codes are the same?

3. Code utilization: In vector quantization, there's often an issue of codebook collapse, where some codes in the codebook are not used, leading to suboptimal performance. For instance, models like VQ-GAN [4] may have a codebook size of 16384 but only use 973 codes written in table 3 of [5], resulting in very low code usage. How does the proposed method perform in terms of code utilization, and does the use of mutual information impact this metric? If mutual information does have an impact, it would highlight the significance of this research.

4. Reason for adopting Gumbel-softmax: There are several methods to construct a vector quantization bottleneck, such as updating via back-propagation as in the original VQ-VAE [1], using an exponential moving average to update the codebook, or using Gumbel-softmax as in this paper. What is the specific reason for choosing Gumbel-softmax in this research? The question arises from personal experience, where a vector quantization module operating with an exponential moving average outperformed one using Gumbel-softmax.


[1] Neural Discrete Representation Learning, NeurIPS 17
[2] Optimized Product Quantization, TPAMI 13
[3] Autoregressive Image Generation using Residual Quantization, CVPR 22
[4] Taming Transformers for High-Resolution Image Synthesis, CVPR 21
[5] Locally Hierarchical Auto-Regressive Modeling for Image Generation, NeurIPS 22